# Cross-species alignment along the chronological axis reveals evolutionary effect on structural development of the human brain

Yue Li[1,2], Qinyao Sun[2], Shunli Zhu[2], Congying Chu[3]*, Jiaojian Wang[1,4]*

[1]State Key Laboratory of Primate Biomedical Research, Institute of Primate Translational Medicine, Kunming University of Science and Technology, Kunming, China; [2]School of Life Science and Technology, University of Electronic Science and Technology of China, Chengdu, China; [3]Brainnetome Center & National Laboratory of Pattern Recognition, Institute of Automation, Chinese Academy of Sciences, Beijing, China; [4]Yunnan Key Laboratory of Primate Biomedical Research, Kunming, China

*For correspondence:
chucongying@gmail.com (CC);
jiaojianwang@uestc.edu.cn (JW)

Competing interest: The authors declare that no competing interests exist.

**Abstract** Disentangling the evolution mysteries of the human brain has always been an imperative endeavor in neuroscience. Although many previous comparative studies revealed genetic, brain structural and connectivity distinctness between human and other nonhuman primates, the brain evolutional mechanism is still largely unclear. Here, we proposed to embed the brain anatomy of human and macaque in the developmental chronological axis to construct cross-species predictive model to quantitatively characterize brain evolution using two large public human and macaque datasets. We observed that applying the trained models within-species could well predict the chronological age. Interestingly, we found the model trained in macaque showed a higher accuracy in predicting the chronological age of human than the model trained in human in predicting the chronological age of macaque. The cross-application of the trained model introduced an individual brain cross-species age gap index to quantify the cross-species discrepancy along the temporal axis of brain development and was found to be associated with the behavioral performance in visual acuity test and picture vocabulary test in human. Taken together, our study situated the cross-species brain development along the chronological axis, which highlighted the disproportionately anatomical development in human brain to extend our understanding of the potential evolutionary effects.

## Editor's evaluation

This important study compared the brain development trajectories of humans and macaque monkeys to quantify different evolutionary effects of convergent and divergent neural pathways between the two species. The cross-species evidence is solid, based on brain age prediction models that were carefully developed by using public MRI datasets of both humans and macaque monkeys. The findings will be of interest to neuroscientists, developmental biologists, and evolutionary biologists.

## Introduction

Understanding the evolutionary effects on the neural system is a fundamental objective in comparative neuroscience. Compared with nonhuman primates (NHPs), the human brain has evolved

dramatically to support the human unique specialized abilities such as language, thinking, sociality, and other high-order cognitive functions (*Saxe, 2006*). Due to higher similarities in brain homology, analogous socio-emotional behaviors, and genetics, the NHPs, particularly rhesus macaques, provide an excellent surrogate for investigating the brain evolution of the human lineage and its relationship to pathophysiology (*Gray and Barnes, 2019*; *Howell et al., 2019*; *Phillips et al., 2014*). Unraveling evolutionary adaptations of the human brain is not only crucial to understand the neural substrates of human unique cognitive functions but also contributes to uncovering pathological mechanisms of the human brain disorders and developing effective therapeutic strategies (*Feng et al., 2020*; *Nelson and Winslow, 2009*; *Thiebaut de Schotten et al., 2019*).

Previous comparative studies have found that primate evolution has remodeled the brain architectures in structures, functions, and wiring patterns. With the development of macroscopic MRI technique, comparative neuroimaging could directly evaluate brain structures, functions, and connectivity differences between humans and NHPs. The structural MRI revealed increased cortex expansion and hemispherical asymmetry in humans compared to NHPs (*Alexander et al., 2001*; *Donahue et al., 2018*; *Hill et al., 2010*; *Leroy et al., 2015*; *Marie et al., 2018*). In addition, human-specific functional brain areas or networks responsible for decision-making and attention emerging during human evolution were reported (*Mantini et al., 2013*; *Neubert et al., 2014*; *Patel et al., 2015*). Advances in diffusion tensor imaging allow direct comparisons of microstructure of fiber tracts, anatomical connectivity strength between brain areas and complex network organization between humans and NHPs. The increased myelination and fiber connections of arcuate fasciculus (AF) for language processing have been consistently found in humans compared to NHPs (*Balezeau et al., 2020*; *Eichert et al., 2019*; *Rilling et al., 2008*; *Sierpowska et al., 2022*). Complex network analysis based on macroscale structural connectivity network revealed different network hubs, and unique wiring properties of the human brain (*Goulas et al., 2014*; *Li et al., 2013*). Recently, we mapped the evolutionary and developmental connectivity atlas of language areas and revealed increased functional balance, amplitude of low-frequency fluctuations, functional integration, functional couplings, and better myelination of dorsal and ventral white matter language pathways in humans compared to macaques (*Cheng et al., 2021*; *Wang et al., 2020*). While these studies offer ample evidence of brain differences between humans and NHPs, it is important to acknowledge that quantifying evolutionary differences through spatial alignment comparisons may partially manifest the evolutionary effects on the human brain, considering the intrinsic evolutionary divergence between humans and macaques about 25 million years ago.

Except for the spatial alignment between the cross-species brains, mounting evidence has demonstrated the existence of brain development along the temporal axis for both humans and macaques in early life. Studies on early human brain development have shown that the sensory cortex tends to develop more quickly than the association cortex, and that language processing specialization occurs in the left hemisphere even during infancy (*Dubois et al., 2014*; *Paus et al., 2001*). Furthermore, cognitive control, emotional processing, and motivation are interdependent and mature during adolescence (*Christakou, 2014*; *Cunningham et al., 2002*; *Kilford et al., 2016*). Additionally, the cortical surface area expands during puberty in rhesus macaques (*Ronan et al., 2014*). Upon the phenomenon of the coexistence of brain development, it raises the question of whether we can conduct the cross-species comparison of the brain by comparing the phases of brain development. In other words, it would be necessary to establish a comparable relationship between brain anatomy and the phases of brain development. By doing so, it would add to our understanding of the (1) differences in the conserved and dominant human and macaque brain structure, function, and networks at corresponding developmental stages; and (2) directly quantify the disproportionate anatomical development and evolution in human and macaque brains at specific developmental stages.

To address the current research bottlenecks of comparative neuroscience lacking anaylsis of human and macaque developmental data along a chronological axis, we here propose a machine learning-based cross-species prediction model, that is, brain structure-based cross-species age prediction model, to quantitatively characterize brain evolutionary pattern along the temporal axis compared to traditional comparative statistical analyses. First, the gray matter volume (GMV) and microstructure of white matter tracts (fractional anisotropy (FA), mean diffusion (MD), axial diffusion (AD), and radial diffusion (RD)) were taken as features to train the prediction models and were used to predict ages for human and macaque, respectively. Then, the trained human and macaque prediction models were

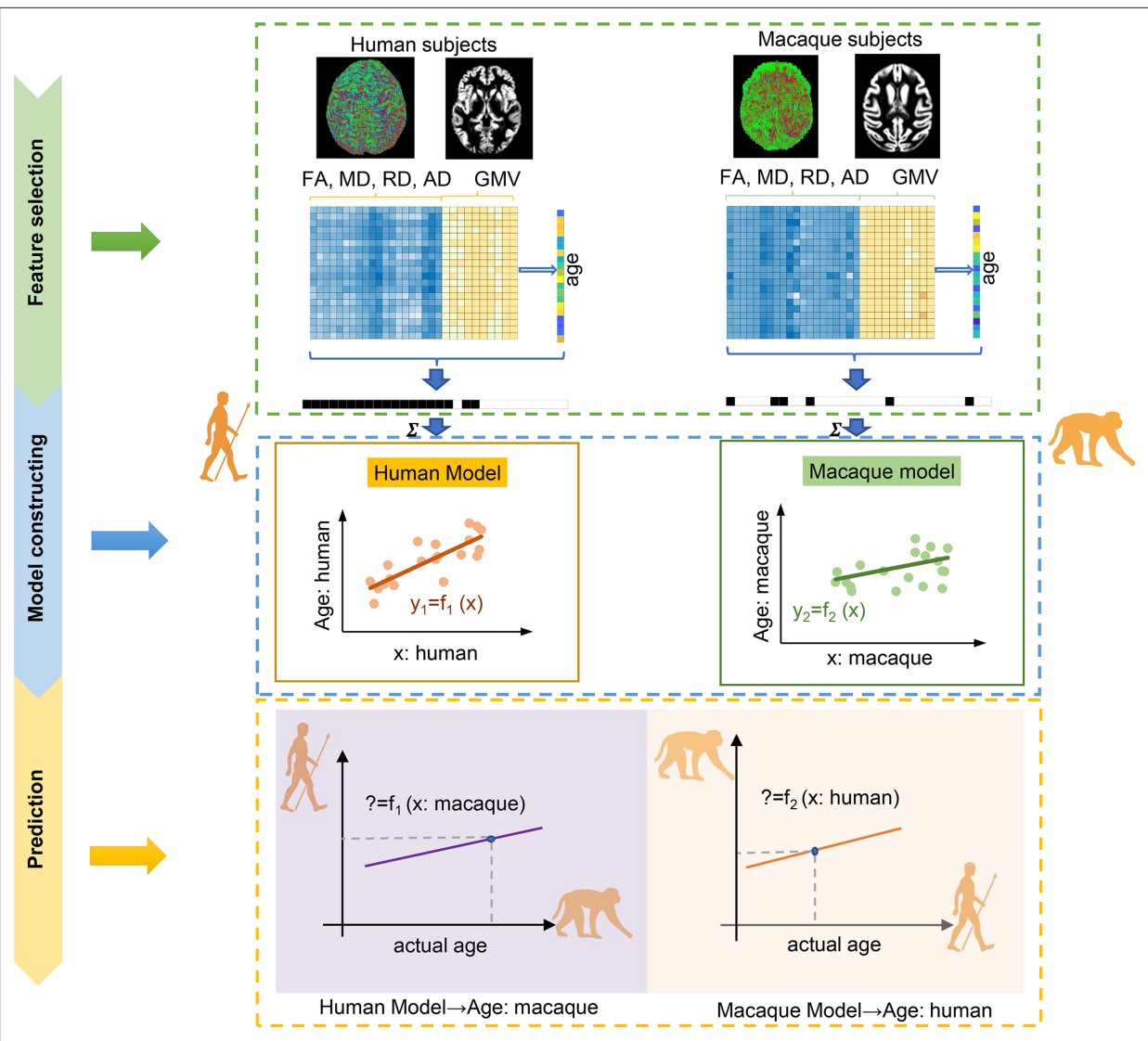

**Figure 1.** The flowchart for intra- and inter-/cross-species prediction using brain structure-based cross-species age prediction model. Feature selection: the features with p<0.01 from Pearson's correlation between features consisted of gray matter volume (GMV), fractional anisotropy (FA), mean diffusion (MD), radial diffusion (RD), and axial diffusion (AD), and ages were preliminarily selected. The ultimate features were selected using two criteria: common features and minimum mean absolute error (MAE) across 100 times repetitions. Model construction: using the selected features, tenfold cross-validation with nine-tenths was used to train the model. Prediction: tenfold cross-validation with one-tenth was used for prediction.

separately applied to predict the ages of macaque and human. Next, based on the macaque prediction model to predict human ages, we proposed a new concept of the brain cross-species age gap (BCAP) to quantify evolutionary differences. Finally, Pearson's correlations were calculated to identify BCAP-associated brain areas, white matter tracts, and behavioral phenotypes. The details of the pipeline of this study are shown in *Figure 1*.

## Results

### Intra- and inter-/cross-species age prediction

The human and macaque brain ages were predicted using the brain structure-based cross-species age prediction model. First, we used 62 macaque features and 225 human features that were present in all the 100 times iterations to train the macaque and human prediction models, respectively (for all the features, see *Figure 2—source data 1 and 2*). Using the trained models, we found that the macaque prediction model can well predict macaque ages ($R = 0.5729$, $p<0.001$, mean absolute

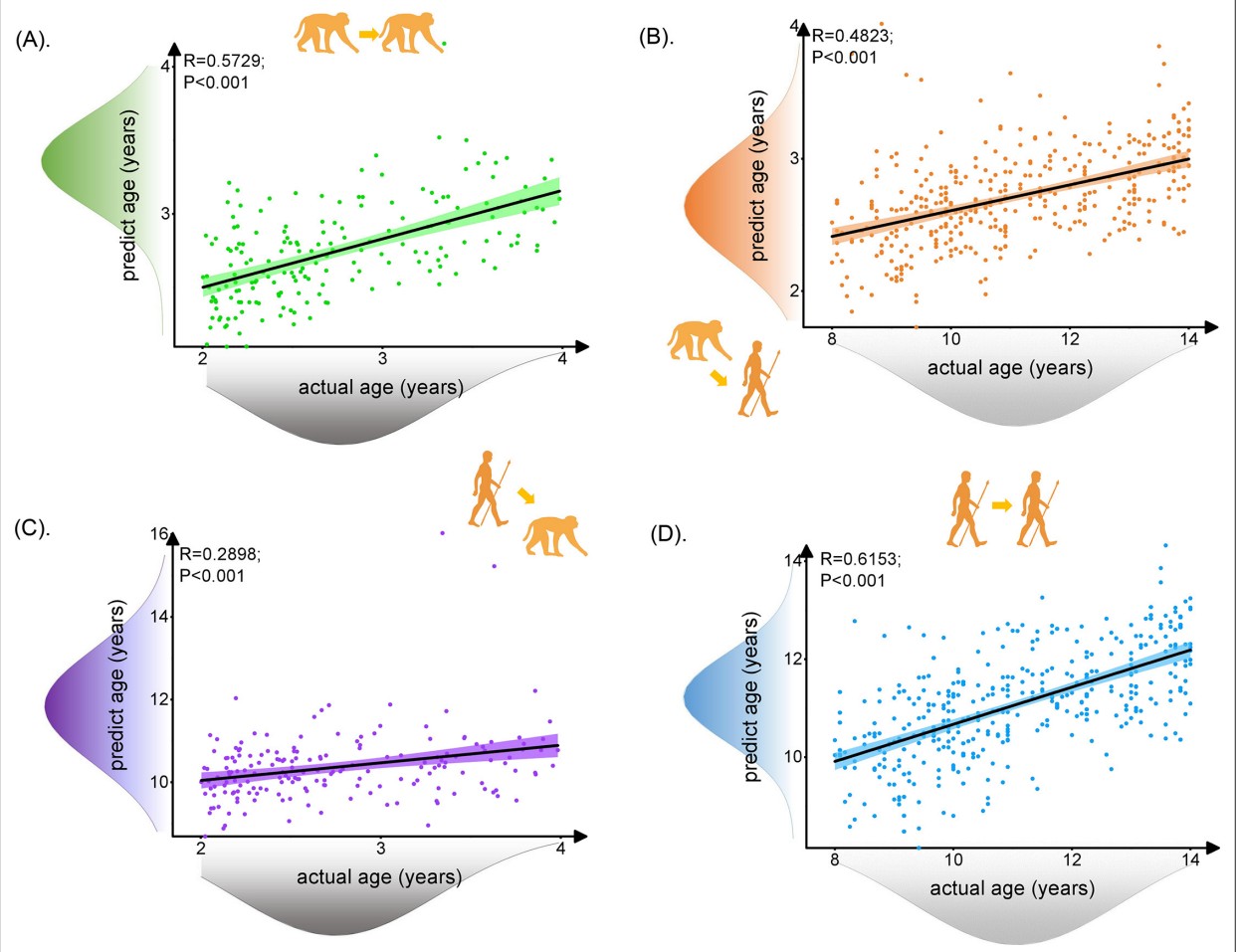

**Figure 2.** The prediction results for intra- and inter-/cross-species using brain structure-based cross-species age prediction model. Each dot depicts data from an individual participant. The width of the curve denotes the 95% CI around the linear fitting curve (black line). The prediction model could well predict intra- and inter-species ages. The trained monkey prediction model could better predict human ages than using trained human prediction model to predict macaque ages. (**A**, **D**) Prediction results for intra-species with green for macaque and blue for human. (**B**, **C**) Prediction results for inter-species with orange for predicted human ages using macaque model and purple for predicted macaque ages using the human model.

The online version of this article includes the following source data and figure supplement(s) for figure 2:

**Source data 1.** The information of 62 selected features of macaque.

**Source data 2.** The information of 225 selected features of human.

**Figure supplement 1.** The prediction results of intra- and inter-species using prediction model with 117 selected macaque features and 239 selected human features.

**Figure supplement 1—source data 1.** The information of 117 selected features of macaque.

**Figure supplement 1—source data 2.** The information of 239 selected features of human.

**Figure supplement 2.** The prediction results of intra- and inter-species using prediction model with 62 selected macaque features and 62 selected human features.

**Figure supplement 3.** The prediction results of intra- and inter-species in males and females.

error [MAE] = 0.3758; *Figure 2A*) and the human prediction model can well predict human ages (*R* = 0.6153, p<0.001, MAE = 1.1236; *Figure 2D*). Interestingly, we applied the trained macaque and human prediction models to separately predict human and macaque ages for inter-/cross-species prediction. We found that the trained monkey prediction model could well predict human ages (*R* = 0.4823, p<0.001, MAE = 8.3610; *Figure 2B*), and the trained human prediction model can also predict macaque ages (*R* = 0.2898, p<0.001, MAE = 7.6157; *Figure 2C*). However, we noticed that using the macaque prediction model to predict human ages showed better performance than using the trained

human prediction model to predict macaque ages. In addition, we observed that the MAE values of the inter-species prediction results are much larger than the MAE values of the intra-species for both human and macaque, which may indicate the potential evolutionary difference during development between human and macaque.

To test the impact of different number of features on prediction performance, we first trained the macaque prediction model with 117 features and human prediction model with 239 features selected using the criterion of minimum MAE during model training with 100 repetitions (for all the features, see *Figure 2—figure supplement 1—source data 1 and 2*). We found that the macaque prediction model and human prediction model can well predict macaque ($R = 0.5825$, p<0.001, MAE = 0.3675) and human ages ($R = 0.6039$, p<0.001, MAE = 1.1388), respectively. We also found that the trained monkey prediction model could well predict human ages ($R = 0.4018$, p<0.001, MAE = 7.7185) and the trained human prediction model can also predict macaque ages ($R = 0.3223$, p<0.001, MAE = 7.9514) (*Figure 2—figure supplement 1*).

Using the same top 62 features of macaque and humans, we observed that the macaque prediction model and human prediction model can well predict macaque ($R = 0.5729$, p<0.001, MAE = 0.3760) and human ages ($R = 0.6818$, p<0.001, MAE = 1.0025), respectively. We also observed that the trained macaque prediction model could well predict human ages ($R = 0.4822$, p<0.001, MAE = 8.3606) and the trained human prediction model can also predict macaque ages ($R = 0.2094$, p=0.0047, MAE = 6.1083). Through testing different number of features, we found that both human and macaque prediction models could well predict their corresponding ages. In addition, we also observed a good performance for the inter-/cross-species prediction. Consistently, we showed that the trained macaque prediction model predicting human ages outperformed the trained human prediction model predicting macaque ages even using different number of features for inter-/cross-species prediction (*Figure 2—figure supplement 2*).

Finally, to explore whether sex affects the prediction model, we separated both human and macaque participants into male and female groups for intra- and inter-species prediction. The predicted results showed similar pattern with that predicted by combining male and female into one group (*Figure 2— figure supplement 3*).

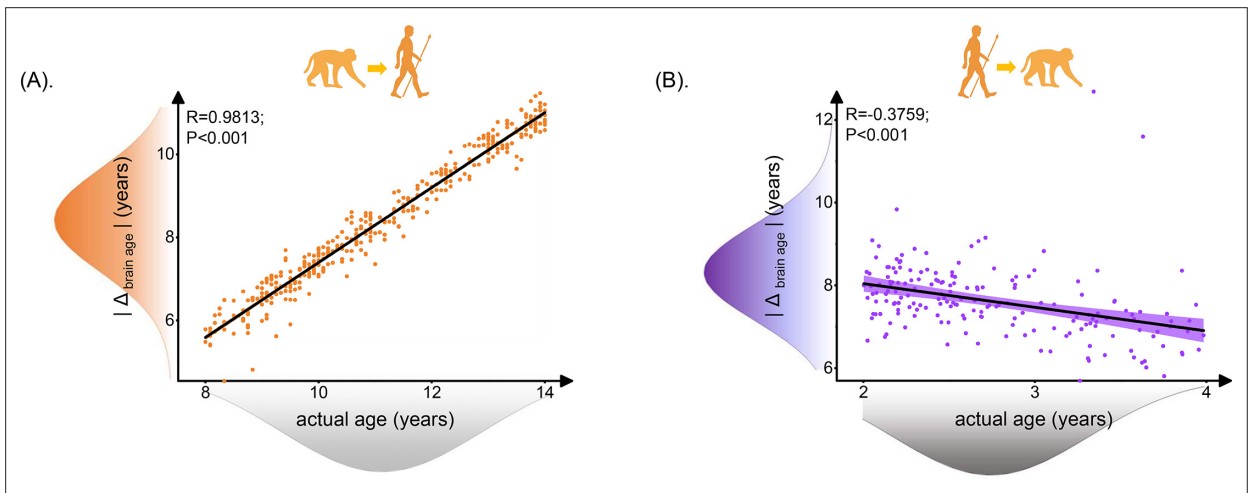

**Figure 3.** Relationship between brain age gap ($|\Delta_{\text{brain age}}|$) and actual ages in human. Each dot depicts data from an individual participant. The width of the curve denotes the 95% CI. The human brain age gap ($|\Delta_{\text{brain age}}|$) is defined as predicted human age using macaque model minus human actual age. The macaque brain age gap ($|\Delta_{\text{brain age}}|$) is defined as predicted macaque age using the human model minus macaque actual age. (**A**) Positive association between $|\Delta_{\text{brain age}}|$ and actual age in human ages was found (Pearson's correlation: $R = 0.9813$, p<0.001, MAE = 2.7120). (**B**) Negative association between $|\Delta_{\text{brain age}}|$ and actual ages in macaque was found (Pearson's correlation: $R = -0.3759$, p<0.001, MAE = 4.8697). MAE: mean absolute error.

The online version of this article includes the following figure supplement(s) for figure 3:

**Figure supplement 1.** Relationship between brain age gap ($|\Delta_{\text{brain age}}|$) and actual ages in human and macaque with 117 selected macaque features and 239 selected human features.

**Figure supplement 2.** Relationship between brain age gap ($|\Delta_{\text{brain age}}|$) and actual ages in human and macaque with 62 selected macaque features and 62 selected human features.

By testing different number of features and sex effects, we observed consistent patterns for intra- and inter-/cross-species age prediction using the prediction model. Importantly, we revealed a stable pattern that the trained macaque prediction model could better predict human ages than that using the trained human prediction model to predict macaque ages.

## Ages effects on the brain age gap of |Δ brain age| for the inter-/cross-species prediction

We evaluated the relationship between actual age and predicted age deviation (|Δ brain age|) for inter-/cross-species prediction. When we used the macaque model to predict human ages, the |Δ brain age| in human showed a positive correlation with the actual age of the human subjects, indicating that the error of using macaque model to predict human age increased while human age increases. On the contrary, when using the human model to predict macaque age, the prediction error of the human model decreased as the macaque age increases (*Figure 3*).

Like above, we also conducted the above correlation analyses between actual ages and predicted age deviation using 117 macaque features and 239 human features and the same top 62 features in both macaque and human; the results obtained with the two sets of features showed similar patterns, demonstrating the stability of the prediction models (*Figure 3—figure supplements 1 and 2*). In addition, to test sex effects, the trends of the correlations between actual age and predicted age deviation are similar to that found using different sets of features, suggesting that the results had

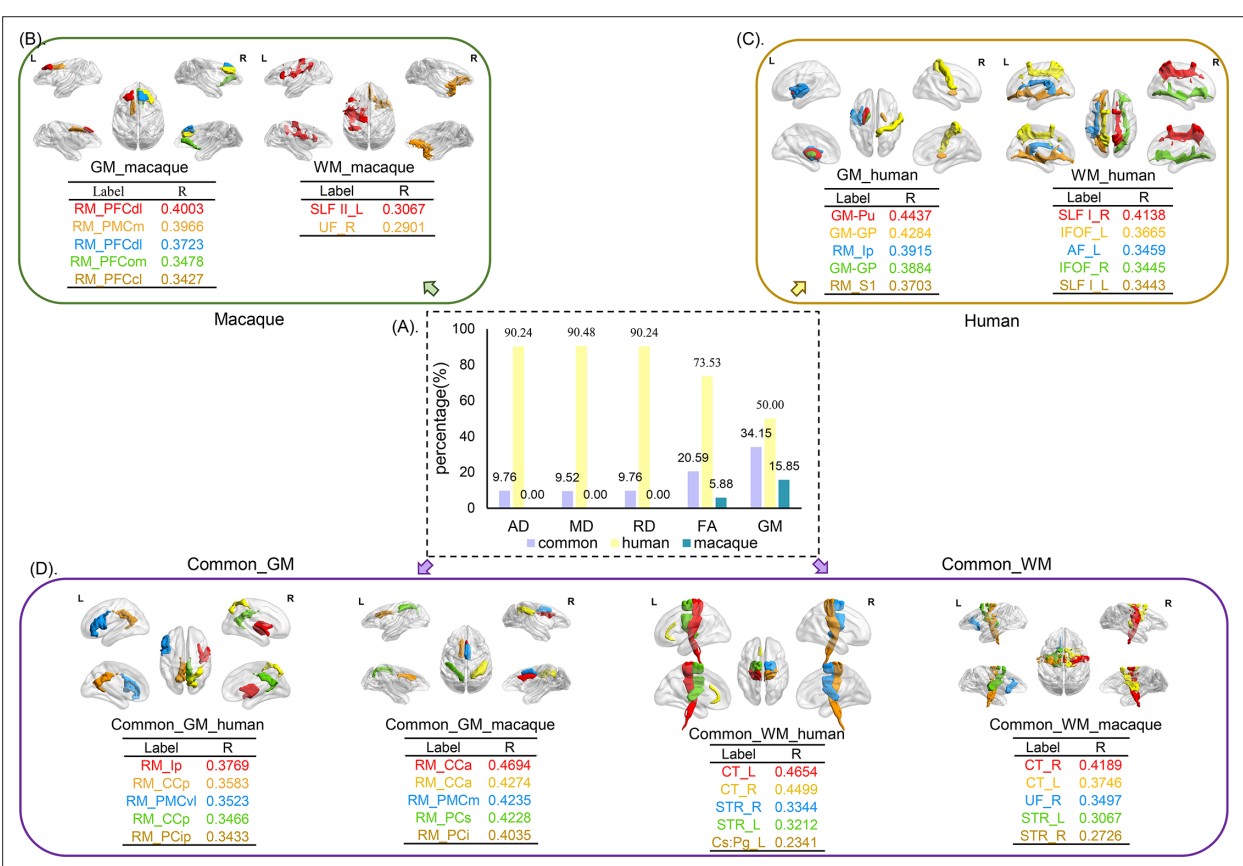

**Figure 4.** The distribution of selected features for prediction. The selected features were analyzed based on five parameters (GMV, FA, MD, RD, and AD) and three groups (human-specific, macaque-specific, and common features in human and macaque). (**A**) The percentage of each group in each parameter. The macaque-specific features are only located in FA and GMV. (**B—D**) The top five features of macaque-specific, human-specific, and common in human and macaque in gray matter and white matter tracts are shown. FA: fractional anisotropy; MD: mean diffusivity; RD: radial diffusivity; AD: axial diffusivity; GMV: gray matter volumes.

The online version of this article includes the following figure supplement(s) for figure 4:

**Figure supplement 1.** The distribution of features with 117 selected macaque features and 239 selected human features.

**Figure supplement 2.** The distribution of features with 62 selected macaque features and 62 selected human features.

no significant difference between sexes (*Figure 2—figure supplement 3*). The decreased prediction accuracy using the macaque prediction model for human ages suggests growing evolutionary differences during human development.

## The brain areas and white matter tracts contributing to prediction

We classified the features for predication in human and macaque into three subtypes (human-specific features, macaque-specific features, and common features of humans and macaques), and the proportion of each subtype of features belonging to GMV, FA, MD, AD, and RD was calculated. The top five brain areas or white matter tracts showing the highest correlations with ages are displayed (*Figure 4*). We observed that macaque-specific features were mainly located in gray matter areas, while human-specific features were mainly distributed in white matter tracts and only small proportions in gray matter areas. The common features of human and macaque have the highest proportion in gray matter areas but also some were located in white matter tracts (*Figure 4A*). For the macaque-specific features, the gray matter features are mainly located in the left dorsolateral prefrontal cortex (PFCdl, $R = 0.4003$), left medial prefrontal cortex (PMCm, $R = 0.3966$), right PFCdl ($R = 0.3723$), right orbitomedial prefrontal cortex (PFCom, $R = 0.3478$), and right centrolateral prefrontal cortex (PFCcl, $R = 0.3427$); and the white matter features are mainly located in the left superior longitudinal fasciculus II (SLF II, $R = 0.3067$) and right uncinate fasciculus (UF, $R = 0.2901$) (*Figure 4B*). For the human-specific features, the gray matter features are mainly located in the left putamen (Pu, $R = 0.4437$), right pallidum (GP, $R = 0.4284$), left posterior insula (Ip, $R = 0.3915$), left GP ($R = 0.3884$), and right primary somatosensory cortex (S1, $R = 0.3703$); and the white matter features are mainly located in the right superior longitudinal fasciculus I (SLF I, $R = 0.4138$), left inferior fronto-occipital fasciculus (IFOF, $R = 0.3665$), left AF ($R = 0.3459$), right IFOF ($R = 0.3445$), and left SLF I ($R = 0.3443$) (*Figure 4C*). The common features of human and macaque are also distributed in gray matter and white matter tracts. The top five brain areas of common gray matter features in human are right Ip ($R = 0.3769$), left posterior cingulate cortex (CCp, $R = 0.3583$), left ventrolateral premotor cortex (PMCvl, $R = 0.3523$), right CCp ($R = 0.3466$), and right intraparietal cortex (PCip, $R = 0.3433$); while the top five brain areas of common gray matter features in macaque are right anterior cingulate cortex (CCa, $R = 0.4694$), left CCa ($R = 0.4274$), right medial premotor cortex (PMCm, $R = 0.4235$), left superior parietal cortex (PCs, $R = 0.4228$), and right inferior parietal cortex (PCi, $R = 0.4035$). The top five common white matter tract features in human are left corticospinal tract (CT, $R = 0.4654$), right CT ($R = 0.4499$), right superior thalamic radiation (STR, $R = 0.3344$), left STR ($R = 0.3212$), and left cingulum subsection: peri-genual (Cs:Pg, $R = 0.2341$); and the top five common white matter tract features in macaque are right CT ($R = 0.4189$), left CT ($R = 0.3746$), right UF ($R = 0.3497$), left SLF II ($R = 0.3067$), and left STR ($R = 0.3046$) (*Figure 4D*).

Meanwhile, using 117 macaque features and 239 human features or the same top 62 features of human and macaque for intra- and inter-/cross-species predication, we observed the distribution of the human-specific, macaque-specific, and common features in gray matter brain areas and white matter tracts are similar to the above results obtained with 62 macaque features and 225 human features-derived prediction models. However, most of the macaque-specific features are located in gray matter while only a few human-specific features are located in gray matter. The highly correlated common feature in human is CT while the highly correlated common macaque feature is CCa (*Figure 4—figure supplements 1 and2*).

## The BCAP-associated behavioral phenotypes, gray matter, and white matter tracts

We explored the behavioral significances of BCAP using partial correlation analyses between behavioral scales and BCAP (the information of behavioral phenotypes and their Pearson's partial correlation results with BCAP is provided in *Figure 5—source data 1*). We observed that BCAP shows significant correlations with picture vocabulary test ($R = 0.1588$, $Pp = 0.0323$) and visual acuity test ($R = -2051$, $p=0.0056$) (*Figure 5A*).

Finally, with our proposed BCAP index, we identified evolution-associated brain areas and white matter tracts in human. The first three and last three white matter tracts in FA showing significantly positive and negative correlations with BCAP were left AF ($R = 0.3784$), left optic radiation (OR, $R = 0.3232$), right AF ($R = 0.3035$) and anterior commissure (AC, $R = 0.1215$), right superior longitudinal

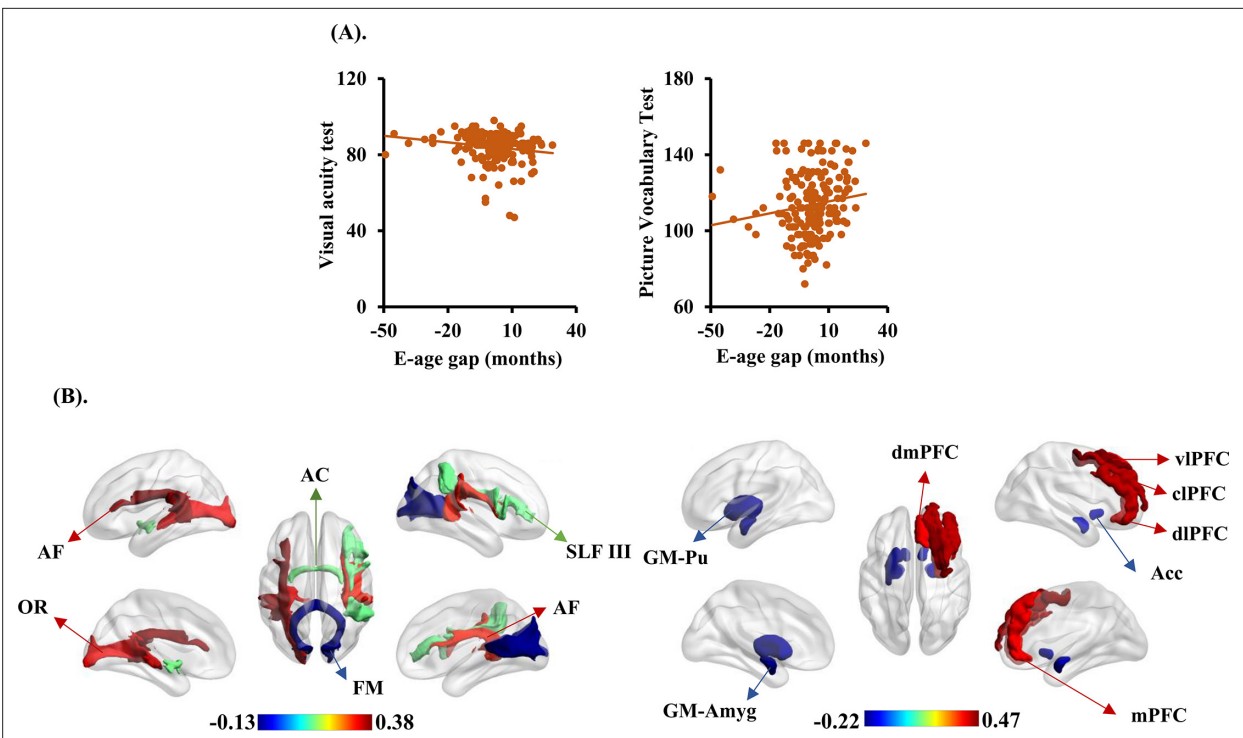

**Figure 5.** The brain cross-species age gap (BCAP) correlations with behavioral phenotypes, gray matter, and white matter tracts. (**A**) Each dot depicts data from an individual participant. Visual acuity test and picture vocabulary test showed negative and positive correlations with BCAP, respectively. (**B**) The top three and last three features associated to BCAP in white matter tracts on the left, and top five and last five features associated to BCAP in gray matter on the right. These white matter tracts and gray matter related to the language pathways, emotion, and higher-order cognitive functions. AF: arcuate fasciculus; OR: optic radiation; AC: anterior commissure; FM: forceps major; SLF III: superior longitudinal fasciculus III; Pu: putamen; Acc: accumbens nucleus; Amyg: amygdala; dmPFC: dorsomedial prefrontal cortex; vlPFC: ventrolateral prefrontal cortex; mPFC: medial prefrontal cortex; dlPFC: dorsolateral prefrontal cortex; clPFC: centrolateral prefrontal cortex.

The online version of this article includes the following source data and figure supplement(s) for figure 5:

**Source data 1.** The information of available behavioral phenotypes and their Pearson's correlation results with BCAP (not all the participants have all the behavioral tests).

**Figure supplement 1.** The top three and last three features associated with E-age gap in mean diffusion (MD), radial diffusion (RD), and axial diffusion (AD).

fasciculus III (SLF III, $R = 0.1166$), and forceps major (FM, $R = –0.1221$); the first five and last five gray matter brain areas showing significantly positive and negative correlations with BCAP were right PFCdl ($R = 0.4685$), right PFCcl ($R = 0.4324$), right ventrolateral prefrontal cortex (PFCvl, $R = 0.4287$), right dorsomedial prefrontal cortex (PFCdm, $R = 0.3915$), and right PMCm ($R = 0.3890$), and right amygdala (Amyg, $R = –0.1759$), right accumbens nucleus (Cd, $R = –0.181$), left accumbens nucleus (Cd, $R = –0.1811$), left Pu ($R = –0.2104$), and left Amyg ($R = –0.2151$) (*Figure 5B*). The first three and last three white matter tracts between BCAP and MD, RD, and AD are shown in in *Figure 5—figure supplement 1*.

## Discussion

In this study, we performed cross-species comparisons of brain development by embedding human and macaque brain anatomy in the perspective of chronological axis applying brain structure-based cross-species age prediction model using multimodal brain imaging data of sMRI and dMRI to quantitatively characterize brain evolutionary patterns. We observed that the prediction model had good performances for intra-/inter- cross-species age predictions regardless of the number of features and sex. Interestingly, we found the imbalanced model performance in inter-species, which was the trained macaque prediction model to predict human ages, had higher accuracy than using the trained

human prediction model to predict macaque ages. Based on the prediction model, we proposed the concept of the BCAP index to quantify the cross-species divergence along the temporal axis during the development and further revealed that BCAP was closely associated with visual acuity test and picture vocabulary test, indicating its potential behavioral significance. Finally, we used BCAP to investigate the evolution along with chronological axis and found that it was mainly associated with language-related white matter pathways and high-order cognitive functions. Taken together, we studied the cross-species brain development along with chronological axis to quantify the disproportionately anatomical development in the human brain, which may provide a new avenue for evolutionary research of the comparative neuroscience.

## Brain structure-based cross-species age prediction model and the BCAP

The traditional comparative studies investigated brain evolution using statistical analyses by spatially aligning the brains between humans and NHPs (*Balezeau et al., 2020*; *Gabi et al., 2016*; *Goulas et al., 2014*; *Rilling, 2014*; *Wang et al., 2020*). Given the intrinsic evolutionary differences of human compared to other NHPs, whether the identified differences could reflect the functional evolution along the time axis remains controversial. In this study, we developed a new approach of brain structure-based cross-species age prediction model and a novel perspective of chronological axis to investigate brain development and evolution. Prediction model could well predict ages for intra- and inter-species regardless of the number of features and sex effect, demonstrating the reliability and stability of the developed method. For the cross-species prediction, we found that the brain age gap (i.e., $|\Delta_{brain\ age}|$) between human actual ages and predicted human age using the macaque prediction model showed positive correlations with human actual age while the macaque brain age gap (i.e., $|\Delta_{brain\ age}|$) between macaque actual ages and predicted macaque ages using human prediction model showed negative correlations with the macaque actual ages, which indicated that the macaque prediction model could better predict the ages of young human participants while the human macaque prediction model could better predict the ages of the older macaques. By analyzing the prediction results corresponding to human and macaque along the timeline, we speculated that the early-developed brain functions are more conservative than the late-developed high-order cognitive functions during species' brain evolution. The prediction errors for human ages gradually increased, indicating that advanced brain connection or activities emerge during development to boost evolutionary differences. Contrarily, the decreased prediction errors for macaque ages using the human model indicated that mature macaque brain may ape young human to some extent. Overall, these findings suggest that the intrinsic evolutionary principles of the brain are effectively captured from a perspective along the timeline with the predictive model we developed.

Although many previous studies have used cortical expansion index to quantitatively characterize brain evolution (*Hill et al., 2010*), the findings about cortex expansion are inconsistent and even contradictory (*Amlien et al., 2016*; *Chaplin et al., 2013*; *Foster et al., 2022*). Thus, there is still lack of a stable index to quantify evolution differences in an intuitive perspective. Based on the cross-species prediction model, we proposed the concept of the BCAP index to quantify brain evolution along with temporal axis. We observed that the BCAP is negatively correlated with visual acuity test while it is positively correlated with picture vocabulary test, indicating that BCAP indeed reflects the evolutionary behavioral phenotypes. In summary, the BCAP may provide an object index to quantify cross-species brain development and evolution along with chronological axis.

## The human- and macaque-specific structural features for intra- and inter-species age prediction

An interesting finding for the prediction features is that the highest proportion of human- or macaque-specific features are primarily located in white matter tracts and gray matter brain areas, respectively. We also observed that disproportionate anatomical development is exhibited in human and macaque brain during the adolescence. The human-specific features are closely related to brain development and mainly include white matter tracts of left AF, inferior fronto-occipital fasciculus (IFOF), and SLF I, which are involved in language and complex motor tasks (*Jacquemont et al., 2023*; *Rilling et al., 2011*). The AF is an important language pathway connecting the key speech production region (Broca's area) in the frontal lobe with the speech comprehension region (Wernicke's area) in the posterior temporal lobe (*Catani and Mesulam, 2008*; *Dick and Tremblay, 2012*; *Rilling et al., 2008*). The

lateralization of AF is reported to be closely associated with language evolution (*Becker et al., 2022*; *Eichert et al., 2019*). The macaque-specific features are mainly distributed in gray matter areas of the prefrontal cortex and motor cortex, which are mainly involved in saccade or visually guided motor and general motor. In addition, some human- or macaque-specific features also were found in gray matter areas and white matter tracts, respectively. The human-specific gray matter features are mainly localized in the insula, basal ganglia, and sensory cortex, which play an important role in emotion, motor, and sensory information control (*Chang et al., 2013*; *Draganski et al., 2008*; *Menon and Uddin, 2010*). The macaque-specific white matter tract features mainly include left SLF II and right UF, which is related to working memory and memory retrieval (*Janelle et al., 2022*; *Makris et al., 2005*; *Papagno et al., 2011*). The analysis of human- and macaque-specific features revealed notable differences in the degree of development of regions associated with language and working memory functions during adolescence between humans and macaques.

## Evolution-related brain areas and white matter tracts

The BCAP was adopted to reveal evolution-related brain areas and white matter tracts and identified similar findings to the human-specific features, indicating disproportionate anatomical brain evolution along the chronological axis. The highly positively correlated white matter tracts are AF, OR, and right SLF II. Language is considered to be the predominant difference between human and other species during evolution. AF is involved in syntax and lexical-semantics processing and is reported to be closely associated with language evolution (*Becker et al., 2022*; *Rilling et al., 2011*). The AF is left lateralization and dorsal predominant in human while symmetrical and ventral predominant in macaques (*Eichert et al., 2019*; *Rilling et al., 2008*). In addition, evolution-related right SLF II connecting parietal cortex and frontal cortex was found. A previous study using task fMRI identified an attention-specific brain area within temporoparietal junction in human (*Patel et al., 2015*). Thus, the evolution-associated right SLF II found in this study may be the underlying white matter pathway for attention.

Moreover, evolution-related gray matter brain areas, including ventrolateral, medial, centrolateral, dorsomedial, and dorsolateral prefrontal cortex, were also uncovered. These frontal cortical areas are involved in high-order cognitive functions, including cognitive control and executive, decision-making, planning, reasoning (*Hiser and Koenigs, 2018*; *Ray and Zald, 2012*; *Salzman and Fusi, 2010*). The prefrontal cortex has been widely reported during evolution with the largest expansion, late myelination, and structural asymmetry in humans compared to other NHPs (*Li et al., 2017*; *Miller et al., 2012*; *Smaers et al., 2013*). In addition to evolution-associated gray matter areas and white matter tracts, we also found some gray matter areas of putamen, amygdala, and subgenual CCa and white matter tracts of forceps minor and OR are negatively correlated with the BCAP. These gray matter areas and white matter tracts mainly participate in motor, visual, and emotion (*Gray et al., 2002*; *Murray, 2007*). The negative correlation may suggest that these brain areas or white matter tracts are more predominant in macaque than in human. Taken together, using the BCAP index, we identify the evolution-associated brain areas and white matter tracts widely reported in previous studies and discovered that the human brain shows a greater proportion of development in language and higher-order cognitive functions through cross-species comparisons of brain development along with chronological axis, which demonstrated that the BCAP is an effective index for future studies to quantify brain evolution between species.

In conclusion, we proposed to embed the brain anatomy of human and macaque in the chronological axis for enabling the cross-species comparison using brain structure-based cross-species age prediction model on brain development. We demonstrated that the prediction model has good performances for cross-species age prediction, and the trained macaque model to predict human ages outperforms the trained human model to predict macaque ages. Moreover, we proposed the index of the BCAP to quantify brain evolution along with temporal axis and identified evolution-associated gray matter areas and white matter tracts consistent with previous reports. The linear regression model developed in this study is preferred over the nonlinear model in terms of parsimony, interpretability, and the underlying relationship between brain structure and age, although the nonlinear model may obtain better accuracy. Importantly, the linear regression model construction using human or macaque brain features allows for a more intuitive comparison to uncover the changing patterns during evolution. However, the existing limitation should be noted regarding the

generalizability of our proposed approach for cross-species brain comparison. Our current model relies on homologous brain atlases, and the lack of comparable atlases for other species restricts its broader applicability. To address this limitation, future research should focus on developing prediction models that do not depend on atlases. For instance, 3D convolutional neural networks could be trained directly on raw MRI data for age prediction. These deep learning models may offer greater flexibility for cross-species applications once the training within species is complete. Such advancements would significantly enhance the model's adaptability and expand its potential for comparative neuroscience studies across a wider range of species. Many studies have reported sex differences in developing human brains (*Hines, 2011*; *Kurth et al., 2021*); however, whether macaque brains have similar sex differences as humans is still unknown. We used the machining learning method for cross-species prediction to quantify brain evolution and the established prediction models are stable even when only using male or female data, which may indicate that the proposed cross-species prediction model has no evolutionary sex difference. Although the stable prediction model can be established in either male or female participants for cross-species prediction, this indeed does not mean that there are no evolutionary sex differences due to the lack of quantitative comparative analysis. In the future, we need to develop a more objective, quantifiable, and stable index for studying sex differences using machining learning methods to further identify sex differences in the evolved brain. In addition, the macaque structural MRI data was preprocessed using SPM8, not the latest version of SPM12, which may affect the analysis accuracy. We will develop a well-established macaque brain structural MRI processing pipeline using CAT12 in SPM12 in future studies. Overall, we introduced a novel protocol and index to investigate and quantitatively characterize brain evolution and discovered disproportionate anatomical development and evolution involving language pathways and cognitive control for human and working memory function for macaque, which may promote the development of comparative neuroscience or neuroimaging.

## Materials and methods

### Human subjects and MRI data acquisition

The used MRI and behavioral phenotypes data were accessed through the public Human Connectome Project-Development (HCP-D: http://www.humanconnectome.org). A total of 370 healthy human subjects (170 males/200 females, age range of 8–14 years, mean and standard deviation of 11.1 ± 1.8 years) with high-quality structure MRI (sMRI) and diffusion MRI (dMRI) were included in this study. All the MRI data were scanned using Siemens 3T Tim Trio MRI scanner with 32-channel head coil and simultaneous multi-slice echo planar imaging sequence. Sagittal 3D T1-weighted images were acquired using the following parameters: time repetition (TR) = 2500 ms, time echo (TE) = 2.22 ms, flip angle = 8°, voxel resolution: $0.8 \times 0.8 \times 0.8$ mm$^3$. The acquisition parameters for dMRI were 185 directions on 2shell of b = 1500 and 3000 s/mm$^2$, along with 28 b = 0 s/mm$^2$, TR/TE = 3230/89.2 ms, flip angle = 78°, voxel resolution: $1.5 \times 1.5 \times 1.5$ mm$^3$, 92 slices. The detailed MRI scanning parameters of the MRI data can be found in a previous study (*Somerville et al., 2018*). In addition, the behavioral phenotypes including behavioral inhibition system and behavioral activation system scale, child behavior checklist, visual acuity test, and picture vocabulary test were also assessed for each individual (for all behavioral phenotype information, see *Figure 5—source data 1*). In order to approximately match the ages of the used macaque data in this study, only 370 subjects with age from 8 to 14 years from HCP-D were finally analyzed. For the HCP-D data, participants provided informed consent (for those under 18 years of age, informed consent was provided by a parent or a legal guardian) at their respective data collection sites and agreed to have their anonymized data shared for research purposes.

### Macaques and MRI data acquisition

All the macaque MRI datasets were scanned at the University of Wisconsin-Madison (UW-Madison). The public structural and diffusion MRI data of 181 macaque monkeys (*Macaca mulatta*, 89 males/92 females, age range from 2 to 4 years, mean and standard deviation = 2.8 ± 0.6 years) were downloaded and used in this study (http://fcon_1000.projects.nitrc.org/indi/PRIME/uwmadison.html), of which 42 macaques were scanned using GE DISCOVERY_MR750 3.0T MRI, and the other 139 macaques were scanned using GE Signa EXCITE 3.0T MRI. All the experimental procedures were

**Table 1.** Demographic information for human and macaque in this study.

| Subjects | Human (8–14 years) | Macaque (2–4 years) |
| --- | --- | --- |
| Number of subjects | 370 | 181 |
| Gender (male:female) | 170/200 | 89/92 |
| Age (mean ± SD) | 11.07 ± 1.75 | 2.75 ± 0.57 |
| Male age (mean ± SD) | 10.89 ± 1.61 | 2.62 ± 0.51 |
| Female age (mean ± SD) | 11.23 ± 1.85 | 2.87 ± 0.60 |

approved by the University of Wisconsin Institutional Animal Care and Use Committee in compliance with the Guide for the Care and Use of Laboratory Animals published by the US National Institutes of Health. Before scanning, all the animals were anesthetized with the following procedures: ketamine, medetomidine, and ketamine maintenance anesthesia. Macaques were scanned in the sphinx position, with the nose pointing into the scanner after approximately 30 min from the first ketamine administration. The physiological features, including heart rate and oxygen saturation, were monitored and recorded at a minimum every 15 min during scanning. Heated water bags, bottles, or pads and towels, blankets, and bubble wrap were used to maintain body temperature during imaging. The scanning parameters of dMRI with GE DISCOVERY_MR750 3.0T were 12 directions with b ≈ 1000 s/mm$^2$, TE/TR = 94.3/6100 ms, flip angle = 90°, voxel resolution = 0.5469 × 2.5 × 0.5469 mm$^3$. The structural T1 images scanning parameters with GE DISCOVERY_MR750 3.0T were TR/TE = 11.4/5.412 ms, flip angle = 10°, voxel resolution = 0.2734 × 0.5 × 0.2734 mm$^3$. For GE Signa EXCITE 3.0T MRI scanner, the dMRI scanning parameters were 12 directions with b ≈ 1000 s/mm$^2$, TE/TR = 77.2/10,000 ms, flip angle = 90°, voxel resolution = 0.5469 × 2.5 × 0.5469 mm$^3$. The structural T1 images scanning parameters with GE Signa EXCITE 3.0T MRI were TR/TE = 8.648/1.888 ms, flip angle = 10°, voxel resolution = 0.2734 × 0.5 × 0.2734 mm$^3$. The detailed MRI scanning parameters are available online (http://fcon_1000.projects.nitrc.org/indi/PRIME/uwmadison.html), and the detailed scanning parameters are available in a previous study (*Young et al., 2017*). All the information for human and macaque is presented in *Table 1*.

### Human and macaque structural MRI data preprocessing

The GMV of human and macaque was calculated using voxel-based morphology (VBM) with SPM8 package (http://www.fil.ion.ucl.ac.uk/spm). The VBM analyses for human subjects using a fully automatic procedure included the following steps: the T1 image was first segmented into gray matter, white matter, and cerebrospinal fluid; the segmented images were then transformed to MNI space using high-dimensional DARTEL normalization and were modulated to account for volume changes. For macaque VBM analysis, the main procedures as follows: the skull of the structural T1 image was first manually stripped and then rotated to match the orientation of the INIA19 template (*Rohlfing et al., 2012*), the skull-stripped T1 image was then automatically segmented into gray matter, white matter, and cerebrospinal fluid and was registered to the INIA19 template using DARTEL-normalization; mean template images of gray matter, white matter, and cerebrospinal fluid were created and each individual GMV was transformed into the new mean gray matter template image. The details for calculating the human and macaque GMV have been described in our previous studies (*Wang et al., 2018*; *Wang et al., 2017*).

### Human and macaque dMRI data preprocessing

The dMRI data for both human and macaque were preprocessed using the FSL software (http://www.fmrib.ox.ac.uk/fsl). The linear portion of eddy currents and head motions of the diffusion MRI were corrected by registering all images to the b0 image using affine transformation to minimize gradient coil eddy current distortions using *eddy_correct* function. Next, the indices of FA, MD, AD, and RD were calculated to quantify the white matter microstructural integrity and/or myelination. Finally, all the FA, MD, AD, and RD maps were registered to template images (MNI152 template for human and INIA19 template for macaque) for further analyses.

## Harmonization for GMV and DTI-derived indices

Given that the differences in imaging protocols, scanning parameters, and scanner manufacturers may affect the reliability of MRI-derived measurements (*Alexander et al., 2001*; *Correia et al., 2009*; *Giannelli et al., 2009*; *Zhan et al., 2010*), we conducted harmonization processing on macaque datasets to ensure the consistency for the following analyses. The harmonization of macaque datasets was performed using the ComBat method, which has been widely applied in previous studies to mitigate bias induced by scanning differences (*Eshaghzadeh Torbati et al., 2021*; *Fortin et al., 2017*; *Johnson et al., 2007*). The harmonization was separately applied to the voxel-wise grey matter volume and the DTI-derived indexes (i.e., FA, MD, and RD) with the covariates, including gender and site using ComBat toolkit in MATLAB Verison 2019b (*Fortin et al., 2017*; *Fortin, 2021*; https://github.com/Jfortin1/ComBatHarmonization).

## Brain structure-based cross-species age prediction model

In this study, we developed a brain structure-based cross-species age prediction model to quantitatively characterize evolution patterns. The indices of GMV, FA, MD, AD, and RD were used as features to construct prediction models for humans and macaques. For cross-species prediction, we used homogeneous brain gray matter and white matter atlases to extract the same features for human and macaque. The brain gray matter and white matter were parcellated into 92 subregions or 42 sub-tracts in both human and macaque using regional map (RM) atlas and XTRACT cross-species atlas, respectively (*Bezgin et al., 2017*; *Warrington et al., 2020*). For each human subject or macaque, a total of 260 features (168 features for white matter tracts [FA, MD, AD, RD for 42 tracts] and 92 features for GMV) were obtained. A linear regression model was adopted for intra- and inter-species age prediction. The linear regression model was built including the following three main steps: (1) feature selection: a total of two steps are required to extract the final features. The first step is preliminary extraction. First, all the human or macaque participants were divided into tenfold and ninefold and used for model training and onefold for model test. The preliminary features were chosen by identifying the significantly age-associated features with p<0.01 while calculating Pearson's correlation coefficients between all the 260 features and actual ages of the ninefold subjects. This process was repeated 100 times. Since we obtained not exactly the same preliminary features each time, we thus further analyzed the preliminary features using two methods to determine the final features: common features and minimum mean absolute error (min MAE). Common features are the preliminary features that were selected in all the 100 times during preliminary model training. The min MAE features were the preliminary features with the smallest MAE value during the 100 times model test for predicting age. After the above feature selections, we obtained two sets of features: 62 macaque features and 225 human features (common features) and 117 macaque features and 239 human features (min MAE). In addition, to further exclude the influences of unequal number of features in human and macaque, we also selected the first 62 features in human and macaque to test the model prediction performances. (2) Model construction: we conducted age prediction with linear model using tenfold cross-validation based on the selected features for human and macaque separately. The linear model parameters are obtained using the training set data and applied to the test set for prediction. The above process is also repeated 100 times. (3) Prediction: with the above results, we obtained the optimal linear prediction models for human and macaque. Next, we performed intra-species and inter-species brain age prediction, that is, human model-predicted human age, human model-predicted macaque age, macaque model-predicted macaque age, and macaque model-predicted human age. Three sets of features (62 macaque features and 225 human features; 117 macaque features and 239 human features; 62 macaque features and 62 human features) were used to test the prediction models for cross-validation and to exclude effects of different number of features in human and macaque. In the main text, we showed the results of brain age prediction, brain developmental, and evolutional analyses based on common features and the results obtained using the other two types of features are shown in the supplementary materials. The prediction performances were evaluated by calculating the Pearson's correlation and MAE between actual ages and predicted ages.

To quantify potential evolutionary differences for the inter-/cross-species prediction models, Pearson's correlation coefficients between the actual ages and the brain age gap ($|\Delta_{\text{brain age}}|$) defined as absolute value of actual age minus predicted age were calculated for both human and macaque.

To determine the feature distributions of the prediction models in human and macaque, we classify the used features of different measures (FA, MD, AD, RD, GMV) into shared/common, human-specific, and macaque-specific features. For each measure, the percentage of the three different types of features was computed. In addition, for each type of the features, the top five brain areas or white matter tracts and the corresponding correlation coefficients with ages are shown.

In addition, to test whether sex affects prediction results, we divided both human and macaque participants into male and female groups. The prediction models were separately trained in males and females and were applied to predict the ages of the corresponding group's subjects. For inter-/cross-species prediction, the trained human or macaque prediction models in males or females were applied to predict the macaque or human ages in males or females, respectively.

## The BCAP-associated brain areas, white matter tracts, and behavioral phenotypes

To better quantitatively characterize brain evolution, we proposed the concept of the BCAP defined as percentile rank differences, which has been demonstrated to be easier to interpret a score or a data point within a data set (*Crawford et al., 2009*). To calculate BCAP, each human subject's predicted age using human prediction model (denoted as P-age$_{human-human}$) was first obtained. Then, the same human subject's predicted age using monkey prediction model (denoted as P-age$_{monkey-human}$) was also calculated. Next, the percentile rank of P-age$_{human-human}$ and P-age$_{monkey-human}$ was computed across all the subjects' predicted P-age$_{human-human}$ and P-age$_{monkey-human}$, respectively. Finally, the BCAP was defined as the percentile of P-age$_{human-human}$ minus the percentile of the P-age$_{monkey-human}$.

To test whether BCAP could reflect behavioral phenotypes, Pearson's partial correlation analyses between BCAP and 40 behavioral phenotypes, including Early Adolescent Temperament Questionnaire, Edinburgh Handedness Questionnaire, Autism Spectrum Rating Scales, Behavioral Inhibition System, and Behavioral Activation System Scale, across all the human subjects were performed with actual ages as covariates. The significant level was determined using false family discovery (FDR) method with p<0.05.

Finally, with the obtained BCAP for each human subject, Pearson's correlation analyses between BCAP and GMV of each brain area, FA, MD, AD, and RD values of each white matter tract across all the human subjects were performed with actual ages as covariates to identify evolution-associated brain areas and white matter tracts. The significantly associated brain areas and white matter tracts were determined using the FDR method with p<0.05. The top three white matter tracts in FA and the top five gray matter brain areas showing the highest and lowest correlation values with ages were shown.

## Acknowledgements

This study was supported by the Science and Technology Innovation 2030-Brain Science and Brain-Inspired Intelligence Project (2021ZD0200900), National Natural Science Foundation (62176044), Natural Science Foundation of Yunnan Province (major basic research project: grant number: 202102AA100053), and Yunnan Fundamental Research Projects (202201BE070001-004).

## Additional information

### Funding

| Funder | Grant reference number | Author |
| --- | --- | --- |
| National Natural Science Foundation of China | 62176044 | Jiaojian Wang |
| Science and Technology Innovation 2030-Brain Science | 2021ZD0200900 | Jiaojian Wang |
| Brain-Inspired Intelligence Project | 2021ZD0200900 | Jiaojian Wang |

| Funder | Grant reference number | Author |
|--------|------------------------|--------|
| Natural Science Foundation of Yunnan Province | 202102AA100053 | Jiaojian Wang |
| Yunnan Fundamental Research Projects | 202201BE070001-004 | Jiaojian Wang |

The funders had no role in study design, data collection and interpretation, or the decision to submit the work for publication.

### Author contributions

Yue Li, Data curation, Writing – original draft, Writing – review and editing; Qinyao Sun, Shunli Zhu, Data curation; Congying Chu, Jiaojian Wang, Conceptualization, Writing – review and editing

### Author ORCIDs

Congying Chu (ID) https://orcid.org/0000-0002-7808-8896
Jiaojian Wang (ID) https://orcid.org/0000-0002-0421-5709

### Ethics

Our used datasets were public data for free downloading and all the datasets were acquired with the approve of ethics committee.For the HCP-D data, participants were provided informed consent (for those under 18 years of age, informed consents were provided for a parent or legal guardian) at their respective data collection sites and have agreed to have their anonymized data shared for research purposes.

All the experimental procedures were approved by the University of Wisconsin Institutional Animal Care and Use Committee in compliance with the Guide for the Care and Use of Laboratory Animals published by the US National Institutes of Health.

### Decision letter and Author response

Decision letter https://doi.org/10.7554/eLife.96020.sa1
Author response https://doi.org/10.7554/eLife.96020.sa2

## Additional files

### Supplementary files

• MDAR checklist

### Data availability

The datasets analyzed during the current study are available at https://www.humanconnectome.org, https://fcon_1000.projects.nitrc.org/. All data generated or analyzed during this study are included in the manuscript and supporting files. The results maps from this study are available at https://github.com/kustlpbr/JiaojianWang-Lab/tree/master/YueLi2024/E_age_gap (copy archived at *JiaojianWang-Lab, 2024*).

The following previously published dataset was used:

| Author(s) | Year | Dataset title | Dataset URL | Database and Identifier |
|-----------|------|---------------|-------------|--------------------------|
| Young JT, Shi Y, Niethammer M, Grauer M, Coe CL, Lubach GR, Davis B, Budin F, Knickmeyer RC, Alexander AL, Styner MA | 2017 | UW-Madison Rhesus MRI dataset | http://fcon_1000.projects.nitrc.org/indi/PRIME/uwmadison.html | PRIMatE Data Exchange, uwmadison |

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
