## [Editor Report]

This important study compared the brain development trajectories of humans and macaque monkeys to quantify different evolutionary effects of convergent and divergent neural pathways between the two species. The cross-species evidence is solid, based on brain age prediction models that were carefully developed by using public MRI datasets of both humans and macaque monkeys. The findings will be of interest to neuroscientists, developmental biologists, and evolutionary biologists.

---

## [Decision Letter]

**Decision letter after peer review:**

Thank you for submitting your article "Cross-species alignment along the chronological axis reveals evolutionary effect on structural development of human brain" for consideration by *eLife*. Your article has been reviewed by 3 peer reviewers, and the evaluation has been overseen by a Reviewing Editor and Yanchao Bi as the Senior Editor.

Essential revisions (for the authors):

While the reviewers were in general positive about the research and the paper, a number of issues were raised. Below are a few main points, which are considered essential revisions and the authors should address them point-by-point before we could recommend acceptance of the paper.

1) Clearly state the theoretical background and research questions.

2) Provide more detailed information and justification on methods. E.g., data sources; the specific implementation and validation of the Combat harmonization process; the training and testing approaches used to evaluate the age prediction models; the algorithm of the prediction model; why SPM 8 was used; the behavioral phenotypes used to test BCAP, and their correlation indices, and describe what the tests (e.g. Behavioral Inhibition System and Behavioral Activation System Scale, Child Behavior Checklist, etc.) involved.

3) Discuss potential confounding factors such as age and gender/sex differences.

4) Discuss limitations, e.g., the current study relied heavily on the use of homogenous brain gray matter and white matter atlases to conduct the cross-species analyses. Also, although the authors named their new method a "cross-species" model, the current study only focused on the prediction between humans and macaques. It would be better to discuss whether their method can also generalize to cross-species examination of other species.

*Reviewer #2 (Recommendations for the authors):*

It is recommended that the authors explore different prediction models for different species. Maybe macaques are suitable for linear prediction models, and humans are suitable for nonlinear prediction models.

*Reviewer #3 (Recommendations for the authors):*

1) Please start by stating that the data for this study came from two large public databases (HCP and UW Madison Rhesus MRI) as it will save time to those readers familiar with the projects. Also, when stating informed consent for human MRI, refer to the respective universities' IRB.

2) Justify why SPM 8 (from 2009) was used for the preprocessing in a 2023/4 study when the newer version, SPM 12 has been available since at least 2014.

3) Provide a brief description of the ComBat method for harmonization of data from different scanners, whether there are other methods available and why this was the most suitable one.

4) Please use references to figure 4 when describing the set of features used to construct the models.

5) Please provide a list of the 40 behavioral phenotypes used to test BCAP, and their correlation indices, and describe what the tests (e.g. Behavioral Inhibition System and Behavioral Activation System Scale, Child Behavior Checklist, etc.) involved.

---

## [Author Response]

Essential revisions (for the authors):Reviewer #3 (Recommendations for the authors):1) Please start by stating that the data for this study came from two large public databases (HCP and UW Madison Rhesus MRI) as it will save time to those readers familiar with the projects. Also, when stating informed consent for human MRI, refer to the respective universities' IRB.

Thank you for your suggestion. We have added the public information in the Materials and methods section. For details, please see page 16, lines 13-14 and page 17, lines 8-13 with green bold font.

Regarding the IRB, because HCP data are a multi-center datasets scanned using same protocols. The informed consent was provided and agreed by each institute but this information was not shared. We have clarified this point in the revised manuscript. For details, please see the page 17, lines 4-6 with green bold font and the following sentences.

For human and macaque public datasets:

On page 16, lines 13-14

“The used MRI and behavioral phenotypes data were accessed through the public Human Connectome Project-Development (HCP-D: http://www.humanconnectome.org).”

On page 17, lines 6-11

“The public structural and diffusion MRI data of 181 macaque monkeys (*Macaca mulatta*, 89 males/92 females, age range from 2-4 years, mean and standard deviation = 2.8 ± 0.6 years) were downloaded and used in this study (http://fcon_1000.projects.nitrc.org/indi/PRIME/uwmadison.html), of which 42 macaques were scanned using GE DISCOVERY_MR750 3.0T MRI, and the other 139 macaques were scanned using GE Signa EXCITE 3.0T MRI”

Regarding the IRB for HCP-D:

“For the HCP-D data, participants were provided informed consent (for those under 18 years of age, informed consents were provided for a parent or legal guardian) at their respective data collection sites and have agreed to have their anonymized data shared for research purposes.”

2) Justify why SPM 8 (from 2009) was used for the preprocessing in a 2023/4 study when the newer version, SPM 12 has been available since at least 2014.

Thank you for your comment. In our study, we used VBM8 toolkit implemented in SPM8 for macaque gray matter volume (GMV) analyses for several reasons. First, SPM8 is more compatible than SPM12 for VBM8 toolkit. Second, in SPM12, the GMV analysis was completed using compatible toolkit of Computational Anatomy Toolbox (CAT12). Although CAT12 toolkit has well-established pipeline for human brain MRI data analysis, there is lack of well-established pipeline for macaque. Third, we have tried to use CAT12 in SPM12 for macaque brain GMV analysis but failed. CAT12 not only needs the tissue probability maps (TPM) of gray matter, white matter and cerebrospinal fluid but also needs the TPM for brain skull to run. But now, there is lack of TPM for brain skull for all the available macaque templates. Thus, we still used our well established VBM8 pipeline for macaque MRI in this study (Wang et al., 2018). We will develop a well-established macaque brain structural MRI processing pipeline using CAT12 in SPM12 in future studies. As you said, this may be a limitation of our study, and we have added this point as a limitation in the revised manuscript. For details, please see page 16, line 4-6 with green bold font as the following sentences.

“In addition, the macaque structural MRI data was preprocessed using SPM8 not the latest version of SPM12, which may affect the analysis accuracy. We will develop a well-established macaque brain structural MRI processing pipeline using CAT12 in SPM12 in future studies.”

3) Provide a brief description of the ComBat method for harmonization of data from different scanners, whether there are other methods available and why this was the most suitable one.

We thank you for your question. In our study, data harmonization with ComBat was used to remove variance contributed by gender and site differences for both the human and the macaque data. In detail, within each species, ComBat was separately applied to the voxel-wise grey matter volume and the DTI-derived indexes (i.e., FA, MD, and RD) with gender and scan sites as covariates. We conducted data harmonization analysis using ComBat toolkit in MATLAB software. Actually, there are a variety of other multisite harmonization methods, like global scaling, functional normalization and RAVEL. Compared with other methods, ComBat can enhance statistical estimation of site effects while preserving the position specificity of multi-site data and improve replicability when the data is age-related and is robust even in small samples (Eshaghzadeh Torbati et al., 2021a; Eshaghzadeh Torbati et al., 2021b; Fortin et al., 2017). Given that Combat has been widely adopted with good performance, we thus also used Combat method for data harmonization in our study.

4) Please use references to figure 4 when describing the set of features used to construct the models.

Thank you for your suggestion. We have added the references to figure 4 when describing the set of features used to construct the models. For details, please see page 6, lines 6-8 with green bold font.

On page 6, lines 6-8

“We showed the top 5 features with the highest correlation with age for each of the three groups of macaque-specific features (Figure 4(B)), human-specific features (Figure 4(C)), and features shared by humans and macaques (Figure 4(D)).”

5) Please provide a list of the 40 behavioral phenotypes used to test BCAP, and their correlation indices, and describe what the tests (e.g. Behavioral Inhibition System and Behavioral Activation System Scale, Child Behavior Checklist, etc.) involved.

Thank you for your suggestion. We conducted Pearson correlation analyses between 40 behavioral phenotypic traits and BCAP to explore whether this index could reflect evolution-related behaviors. Specifically, the BIS/BAS scale (Behavioral Inhibition System and Behavioral Activation System Scale) was used to assess individual motivation towards avoiding unfavorable outcomes and approaching goal-oriented motivations. For example, items in the scale include statements such as "When I am pursuing something I want, I feel excited and energized" and "Even if something bad is about to happen, I rarely feel afraid or nervous." Additionally, the CBCL (Child Behavior Checklist), completed by caregivers, assessed children's social competence and behavioral issues. For instance, it includes questions about the number of friends a child has and the frequency of nervous behaviors. Specific behavioral phenotype tests are online (HCP-D: http://www.humanconnectome.org).

And we added the information of behavioral phenotypes and Pearson’s correlations of BCAP and behavioral scale scores. For details, please see Figure 5-source data 1 in Supplementary Information.

Reference

Eshaghzadeh Torbati, M., Minhas, D. S., Ahmad, G., O'Connor, E. E., Muschelli, J., Laymon, C. M., Yang, Z., Cohen, A. D., Aizenstein, H. J., Klunk, W. E., Christian, B. T., Hwang, S. J., Crainiceanu, C. M., & Tudorascu, D. L. 2021a. A multi-scanner neuroimaging data harmonization using RAVEL and ComBat. *Neuroimage*, 245: 118703.

Eshaghzadeh Torbati, M., Minhas, D. S., Ahmad, G., O’Connor, E. E., Muschelli, J., Laymon, C. M., Yang, Z., Cohen, A. D., Aizenstein, H. J., Klunk, W. E., Christian, B. T., Hwang, S. J., Crainiceanu, C. M., & Tudorascu, D. L. 2021b. A multi-scanner neuroimaging data harmonization using RAVEL and ComBat. *NeuroImage*, 245: 118703.

Fortin, J. P., Parker, D., Tunç, B., Watanabe, T., Elliott, M. A., Ruparel, K., Roalf, D. R., Satterthwaite, T. D., Gur, R. C., Gur, R. E., Schultz, R. T., Verma, R., & Shinohara, R. T. 2017. Harmonization of multi-site diffusion tensor imaging data. *Neuroimage*, 161: 149-170.

Hines, M. 2011. Gender development and the human brain. *Annu Rev Neurosci*, 34: 69-88.

Kurth, F., Gaser, C., & Luders, E. 2021. Development of sex differences in the human brain. *Cogn Neurosci*, 12(3-4): 155-162.

Wang, J., Feng, X., Wu, J., Xie, S., Li, L., Xu, L., Zhang, Y., Ren, X., Hu, Z., Lv, L., Hu, X., & Jiang, T. 2018. Alterations of Gray Matter Volume and White Matter Integrity in Maternal Deprivation Monkeys. *Neuroscience*, 384: 14-20.